# Orthostatic Intolerance in Long-Haul COVID after SARS-CoV-2: A Case-Control Comparison with Post-EBV and Insidious-Onset Myalgic Encephalomyelitis/Chronic Fatigue Syndrome Patients

**DOI:** 10.3390/healthcare10102058

**Published:** 2022-10-17

**Authors:** C. (Linda) M. C. van Campen, Frans C. Visser

**Affiliations:** Stichting Cardiozorg, 1171 JE Badhoevedorp, The Netherlands

**Keywords:** orthostatic intolerance, long-haul COVID, myalgic encephalomyelitis/chronic fatigue syndrome (ME/CFS), tilt testing, cerebral blood flow, post-viral, insidious

## Abstract

Background: As complaints of long-haul COVID patients are similar to those of ME/CFS patients and as orthostatic intolerance (OI) plays an important role in the COVID infection symptomatology, we compared 14 long-haul COVID patients with 14 ME/CFS patients with a post-viral Ebstein-Barr (EBV) onset and 14 ME/CFS patients with an insidious onset of the disease. Methods: In all patients, OI analysis by history taking and OI assessed during a tilt test, as well as cerebral blood flow measurements by extracranial Doppler, and cardiac index measurements by suprasternal Doppler during the tilt test were obtained in all patients. Results: Except for disease duration no differences were found in clinical characteristics. The prevalence of POTS was higher in the long-haul patients (100%) than in post-EBV (43%) and in insidious-onset (50%) patients (*p* = 0.0002). No differences between the three groups were present in the prevalence of OI, heart rate and blood pressure changes, changes in cerebral blood flow or in cardiac index during the tilt test. Conclusion: OI symptomatology and objective abnormalities of OI (abnormal cerebral blood flow and cardiac index reduction during tilt testing) are comparable to those in ME/CFS patients. It indicates that long-haul COVID is essentially the same disease as ME/CFS.

## 1. Introduction

Early in the onset of the SARS-CoV-2 pandemic, it was recognized that complaints persisted in a minority of patients. The presentation of the acute symptoms in people affected with the respiratory virus SARS-CoV-2 (COVID-19) varies greatly [1]. Some people remain asymptomatic, while others have symptoms such as other respiratory viral diseases such as fever, cough, shortness of breath, headache, and a sore throat. Gastrointestinal complaints, complaints of the liver, rheumatological complaints or neurological complaints have also been reported [2,3,4].

In contrast, the set of complaints in the group of patients with persisting complaints is respiratory symptoms, neurological symptoms, fatigue, pain, mental dysfunction, cardiovascular dysfunction, post-exertion symptoms, and cognitive dysfunction. Furthermore, the early onset of orthostatic intolerance symptoms has been described [5,6,7]. The name of this set of persisting complaints has not yet been clearly defined and is called “long” COVID, long-haul COVID, chronic COVID, post-COVID syndrome and even post-acute sequelae of a SARS-CoV-2 infection (PASC) [1,7]. In this study, we use the term “long-haul” COVID. Despite a vast number of studies, the time interval between the onset of the infection and the adjudication of patients to the long-haul COVID group is not established. Intervals vary between 4 weeks and 3 months according to the WHO, CDC, NICE, and AWMF [7]. Interestingly, there does not seem to be a relationship with the severity of the SARS-CoV-2 infection [8].

The symptom characteristics of long-haul COVID resemble those of myalgic encephalomyelitis/chronic fatigue syndrome (ME/CFS) [1,9]. One of the overlapping symptoms is orthostatic intolerance [10]. Orthostatic intolerance is defined as a clinical syndrome in which complaints increase while standing and are relieved by lying down [11]. The orthostatic intolerance is a core criterion of ME/CFS [12]. Especially, the sub form: postural orthostatic tachycardia syndrome (POTS) has been recognized in long-haul COVID [13]. We have quantified the underlying mechanism of orthostatic intolerance, being cerebral hypoperfusion, in patients with ME/CFS using extracranial Doppler during a tilt table test [14]. We showed that in ME/CFS patients cerebral blood flow decreased by 26%, while in healthy volunteers this decrease was on average 7% [14].

In a previous study, we compared long-haul patients with POTS with a matched ME/CFS control group with POTS, with ME/CFS patients with a normal heart rate and blood pressure response during a tilt test and with healthy controls [15]. The prevalence of symptom clusters and the objective signs of orthostatic intolerance (cerebral blood flow reductions during a tilt test) were similar between long-haul COVID patients and the two ME/CFS groups. As the onset of ME/CFS is regarded to be mainly triggered by an Ebstein-Barr virus (EBV) [16,17]; in the present retrospective study, we compared 14 long-haul COVID patients with 14 gender- and age-matched ME/CFS patients whose onset was triggered by EBV. Furthermore, a significant subset of ME/CFS patients who have no clear trigger for the start of the disease: an insidious onset [17]. To explore whether there are differences between the acute, viral onset and the insidious onset, we also included 14 gender- and age-matched ME/CFS patients with an insidious onset of ME/CFS.

## 2. Materials and Methods

### 2.1. Patients

In the period from December 2020 to March 2022, 14 patients with long-haul COVID complaints were investigated because of the suspicion of orthostatic intolerance and/or dysautonomia complaints. In the first 11 patients, there was a clinical suspicion of a SARS-CoV-2 infection, in the last 3 patients the diagnosis was confirmed by a serological test. At the beginning of 2020 (from February to June 2020), serological tests were not conducted or discouraged due to poor availability. The patients without a positive test were included in the study if the clinical diagnosis was made by a pulmonologist, internist, general practitioner or by the public health service. The patients with long-haul COIVD where the disease started before the summer of 2021 were not vaccinated for COVID-19. The 2 patients who contracted the disease after vaccinations started, had been vaccinated fully. All long-haul COVID patients were not hospitalized or treated with oxygen during the acute illness. All patients classified the acute illness as a more or less flu-like disease. All long-haul COVID patients were healthy active subjects without co-morbid diseases. Patients with clear organ damage due to COVID as pulmonary damage resulting from ARDS or severe pneumonia, myocardial infarction or stroke did not visit our outpatient clinic. For the present study 14 ME/CFS patients, age- and gender-matched, were included, where the disease ME/CFS started with an EBV infection. This infection was clinically diagnosed by the general practitioner, and in all cases confirmed by serological examination. Moreover, also 14 ME/CFS patients were included with an insidious onset of the disease, to contrast with the acute onset of the SARS-CoV-2 and the EBV infection. All patients of the three groups underwent a tilt table test with cerebral blood flow measurements for quantification of their orthostatic intolerance. The ME/CFS patients met both the criteria for ME [18] and the criteria for CFS [19], taking the exclusion criteria for ME and CFS into account. In none of the long-haul COVID and ME/CFS patients, there was any other explanation for the symptoms. During the study, patients were not taking medication that could affect heart rate and/or blood pressure. In the long-haul COVID group, one patient used salbutamol occasionally but did not use it in the 2 months before the consultation, 2 patients were on antidepressants and 2 different patients were on antihistamines. Two other patients used vitamin supplements. Finally, prior to the tilt table examination, the orthostatic intolerance complaints in daily life were assessed. On basis of questions about complaints of dizziness, lightheadedness, previous syncope, nausea, sweating, etc., and also on basis of questions about triggers for the development of these complaints, such as standing in a line, showering, etc., the diagnosis of orthostatic intolerance in daily life was made.

The study was conducted in accordance with the Helsinki Declaration. All patients gave written permission for the use of their data. The use of clinical information was approved by the medical ethics committee of the Slotervaart Hospital in Amsterdam (P1736).

### 2.2. Determination of the Severity of the Disease Using the ME Criteria [18]

To classify the clinical severity of ME/CFS, the ME criteria were used. A mild form was defined as an approximately 50% reduction from the pre-disease activity level, a moderate form being mainly housebound and a more than a 50% reduction of the pre-disease activity level. Patients with a severe form were mainly bed-bound. Very severe patients were not included in the study, because they were not able to perform a tilt table test.

### 2.3. Tilt Test Protocol

The methodology used for the tilt table test was previously described [14,20]. In short, all participants were examined in a supine position for 20 min before being tilted to 70 degrees. The average duration of tilting was 10 min. Heart rate and blood pressure were measured with finger plethysmography [21,22]. The changes in heart rate and blood pressure were classified as described by Freeman et al. and Sheldon et al. [23,24]. The classification involved a normal heart rate and blood pressure response, orthostatic hypotension (a decrease of more than 20 mmHg in systolic blood pressure or more than 30 mmHg in case of systolic blood pressure above 160 mmHg) [25], or a diastolic blood pressure reduction of more than 10 mmHg. A sustained increase of at least 30 beats per minute within 10 min of standing, without a significant decrease in blood pressure, was defined as POTS. Syncope involved a temporary loss of consciousness and muscle tone with spontaneous and complete recovery.

### 2.4. Doppler Echocardiography for Stroke Volume and Cardiac Index Measurements

Stroke volume and cardiac index were measured as described before [26]. Time-velocity integral (VTI) images were obtained in a supine position and in the upright position just before tilting back. The VTI of the aorta was measured by a continuous wave Doppler pencil probe connected to a Vivid I ultrasound machine (GE, Hoevelaken, The Netherlands), with the transducer positioned in the suprasternal notch. A maximal Doppler signal was assumed to be a display of the optimal flow. At least 2 images of 6 s were captured. Echo Doppler recordings were stored digitally. The VTI was measured offline by manually tracing the VTI contour of at least 6 heartbeats using the GE EchoPac post-processing software. This was carried out by one researcher (CMCvC). The stroke volume index was calculated using the VTI of the aorta multiplied by the corrected valve area as described earlier [27,28], and divided by the body surface (BSA; DuBois formula) and expressed in mL/m^2^. The stroke volume index of the different heartbeats was averaged. The cardiac index was calculated by multiplying the stroke volume index by the heart rate and was expressed in L/min/m^2^.

### 2.5. Extracranial Doppler for Cerebral Blood Flow Measurements

Measurements were carried out as described earlier [14,20]. The internal carotid and vertebral artery’s Doppler velocities were obtained by one operator (FCV). Frames were recorded in the supine position and in the upright position just before tilting back. The blood flow of the carotid and vertebral arteries was calculated offline by one researcher (CMCvC) who was not familiar with the severity of the disease. Blood flow was calculated for each vessel by multiplying the mean blood flow velocity with the blood vessel surface and was expressed in ml/minute. Blood flow in the individual arteries was calculated in 3–6 heartbeats and the results were averaged. The total cerebral blood flow was calculated by adding the blood flow of the 4 arteries together.

### 2.6. End-Tidal pCO_2_ Measurements

Throughout the test, continuous end-tidal pCO_2_ measurements were obtained by the Nonin Lifesense device (Nonin, Finland), connected with nasal prongs. Data were stored digitally. The averaged P_ET_CO_2_ data of the time interval of the Doppler measurements were taken.

### 2.7. Orthostatic Intolerance Questionnaire during Tilting

The complaints that were asked for were dizziness or lightheadedness, fatigue, muscle weakness, palpitations, shortness of breath, blurred vision, hearing differently, neck and/or shoulder muscle pain, low back pain, pressure or chest pain, concentration problems, sweating, headache, or pressure in the head, tingling and nausea. See for details of the questionnaire the Appendix A.

### 2.8. Statistics

Data were analyzed using GraphPad Prism version 6.05 statistics program (Graphpad software, La Jolla, CA, USA). All continuous data were tested for normal distribution with the D’Agostino and Pearson omnibus normality test and were presented as mean and standard deviation (SD) or as median with interquartile range (IQR) where appropriate. Nominal data were compared with the Chi-square test (a 3 × 2 or 3 × 3 table). Groups were compared using the Student’s *t* test for unpaired data. Between-group comparisons were made by the one-way analysis of variance (ANOVA) or the Kruskal–Wallis test where appropriate. When results were significant, they were further explored using the post-hoc Tukey’s multiple comparison test or Dunn’s test where appropriate. For statistical significance, we chose a conservative *p*-value of <0.01.

## 3. Results

Table 1 shows the clinical characteristics of long-haul COVID and ME/CFS patients. The duration of the disease was significantly longer in both ME/CFS patients groups compared to the long-haul COVID patients (*p* < 0.0001). The other clinical characteristics were not different, including the division between a mild, moderate, and severe form of the disease and the prevalence of orthostatic intolerance symptoms in daily life.

Table 2 shows the hemodynamic results of the tilt table test. POTS was observed in all 14 long-haul COVID patients, 6 patients had POTS and 8 patients showed a normal heart rate and blood pressure pattern in the ME/CFS patients with the EBV trigger, and in the ME/CFS group with an insidious onset 7 patients had POTS and 7 patients showed a normal heart rate and blood pressure pattern (*p* = 0.002). Although more patients with POTS were present in the long-haul COVID patients compared to both ME/CFS patients groups, the heart rates during the upright phase of the tilt test were not significantly different. Blood pressure, stroke volume index, cardiac index and cerebral blood flow supine and upright were not different between long-haul COVID patients and both ME/CFS patient groups.

Figure 1 shows the percentage reduction in stroke volume index, cardiac index and cerebral blood flow in the three groups. There were no significant differences.

Figure 2 shows the number of complaints shortly after the onset of the tilt phase as administered with the standardized questionnaire. In red the percentage of the various complaints in long-haul COVID patients are given, in yellow in ME/CFS patients by EBV and in green ME/CFS with an insidious onset. There is no significant difference in the prevalence of complaints. 

Figure 3 shows examples of cerebral blood flow reduction in long-haul COVID patients in the upright position compared to the supine position. 

Figure 4 shows examples of cerebral blood flow reduction in ME/CFS patients triggered by EBV in the upright position compared to the supine position. 

Figure 5 shows the cerebral blood flow reduction in the upright position compared to the supine position.

## 4. Discussion

Recent research increasingly suggests that the signs and symptoms of long-haul COVID patients are the same as of ME/CFS patients [1,9]. Celi et al. described in a review article the symptomatology of COVID-19 patients in the post-acute phase and in the phase of COVID as a chronic disease [9]. Many of these complaints are complaints that are also present in ME/CFS. Wong and Welzer concluded in a review of 21 long-haul COVID studies that there was a large overlap between long-haul COVID symptoms and ME/CFS symptoms [1].

In the current study, complaints, and the objective abnormalities of orthostatic intolerance in long-haul COVID patients, where the trigger was the SARS-CoV-2 infection, were compared with those of ME/CFS patients with an EBV infection as a trigger as well as with a group of patients with no clear trigger for the start of the disease: an insidious onset. In ME/CFS patients, the triggers for the development of complaints are numerous, which could lead to group heterogeneity. Examples of triggers include infection-related, environmental toxins, stressful incidents, pregnancy, surgeries, trauma, travel, neurological and cardiologic events [17]. Vaccinations or an insidious course are also possible. To improve comparability, we have compared the long-haul COVID patients with ME/CFS patients caused by a single trigger: the Ebstein Barr virus, but also compared it to a group of ME/CFS patients where no clear starting point was present. The main finding is that in long-haul COVID patients, the decrease in cerebral blood flow during a tilt table test is similar to that of ME/CFS patients with a post-viral trigger and is similar to that of patients with an insidious onset. In addition, the complaint pattern of orthostatic intolerance during the tilt is also similar for the three groups. In a previous study, we have shown that in healthy controls, the decrease in cerebral blood flow during a tilt table test was on average 7% and 26% in ME/CFS patients [14]. The abnormalities of the cerebral blood flow decrease of 29% in ME/CFS patients in the current study are similar to the previously found decrease of 26% in 429 ME/CFS patients. Our findings are consistent with the findings of a decrease in cerebral blood flow rates measured with transcranial Doppler in patients with orthostatic intolerance [29,30]. There is one case report that describes the transcranial Doppler measurements, showing brain hypoperfusion in long-haul COVID [31].

Furthermore, this study shows that there is a similar, abnormal decrease in stroke volume index and cardiac index during the standing phase of the tilt table test [32,33]. Finally, it was also found that the orthostatic intolerance complaints in daily life were very similar between long-haul COVID and ME/CFS patients. In a recent publication of ours, we compared the self-reported complaints of 10 long-haul COVID patients with 20 ME/CFS patients [6,15]. We found similar symptom clusters between the long-haul COVID patients and the ME/CFS patients. Thus, the complaints of long-haul COVID patients, the severity of the complaints, the orthostatic intolerance complaints in daily life, the orthostatic intolerance complaints during the tilt table test and the objective abnormalities of the orthostatic intolerance found (the abnormal cerebral blood flow decrease and the abnormal cardiac index decrease) are very similar to those of ME/CFS patients. This makes it likely that long-haul COVID is the same in terms of disease as ME/CFS, where the trigger for its onset has been an infection with the SARS-CoV-2 virus. From a clinical point of view, long-haul COVID patients should be treated the same as ME/CFS patients, focusing on the main complaints such as fatigue, orthostatic intolerance, memory/concentration problems, post-exertional malaise, sleep abnormalities, pain, etc. The symptomatic treatment should depend on the severity of their presenting complaints.

It is striking that POTS was found in all long-haul COVID patients, while this was only 42–50% in the respective ME/CFS patient groups. In a previous study, we found that patients with POTS had a shorter disease duration than patients with a normal heart rate and blood pressure response during the tilt test [34]. In the current study, there was a significant difference in disease duration: median 1 year in the long-haul COVID versus median 11–16 years in the ME/CFS patient groups. One possible explanation for the different prevalence of POTS in long-haul COVID and ME/CFS patients is that with increasing disease duration, the POTS response can be replaced by a lesser increase in heart rate, almost as though it “fades out” and give way to a normal heart rate and blood pressure course. In acute and severe COVID-19 the potential role of excess catecholamines states have been discussed but not properly investigated (gubbi LancetDiabEndocr 2020, Ouyang eur review medic pharmacol science 2021). As one of the mechanisms of POTS is excess sympathetic activation (Vernino 2021), we hypothesize that also in milder COVID-19 patients, initially a high sympathetic drive is present, which disappears over time during recovery of the infection. This recovery then leads to disappearance of POTS. This hypothesis is supported by patients, stating that the palpitations have disappeared over time. The improvement of POTS in a COVID patient was reported by Ocher et al. [35]. The change in hemodynamic profiles over time in long-haul COVID patients needs to be studied further.

Deconditioning as an important cause of POTS is frequently mentioned in the literature [36,37,38]. In the present study of 14 long-haul COVID patients, as well as in our previous publication of 10 patients with POTS [15], complaints of orthostatic intolerance/POTS developed in the early weeks after the onset of the acute infection. None were hospitalized or required mechanical ventilation. Furthermore, all long-haul COVID patients were fit before the onset of the infection and performed physical exercise at least on a weekly basis. Similar observations have been made by others [39,40,41]. This makes it unlikely that the POTS was triggered by deconditioning. Furthermore, in ME/CFS patients we have shown that the presence of postexertional malaise, orthostatic intolerance and the abnormal reduction in cerebral blood flow was also present in patients without deconditioning as assessed by cardiopulmonary exercise testing, where the absence of deconditioning was defined by a percent predicted maximum oxygen consumption of 85% or more [42]. Moreover, we have shown that the abnormal cerebral blood flow reduction was similar in patients with no deconditioning, with mild or severe deconditioning [43,44]. Due to the similarity of long-haul COVID and ME/CFS, exercise therapy should be cautiously given to long-haul COVID patients, as it may also be detrimental [45,46,47].

### Limitations

We excluded patients with different triggers for the development of ME/CFS than an Ebstein Barr virus. Previous research in ME/CFS, which included all triggers for the disease, showed no distinction in demographic characteristics, symptoms, and signs [17]. Thus, the probability that a post-viral trigger will show a different pattern of complaints and objective abnormalities is unlikely. This was confirmed by the similar study results of the group with an insidious start of ME/CFS. None of the long-haul COVID patients developed orthostatic hypotension or syncope or had a normal heart rate and blood pressure pattern in the current study. Large-scale studies, in which it can be expected that these hemodynamic abnormalities do occur in a number of patients, should reveal any differences in complaints and cerebral blood flow abnormalities. Finally, inclusion bias in the referral, because often patients ask for a referral from their GPs after information on social media, may have played a role in the current research.

## 5. Conclusions

The complaints of long-haul COVID patients, the severity of the complaints, the orthostatic intolerance complaints in daily life, the orthostatic intolerance complaints during the tilt table test and the objective abnormalities of the orthostatic intolerance (the abnormal cerebral blood flow decrease and the abnormal cardiac index decrease) are very similar to those of the ME/CFS patients. This makes it very likely that long-haul COVID is the same in terms of disease as ME/CFS, where the trigger for its onset has been an infection with the SARS-CoV-2 virus. The treatment of long-haul COVID should therefore be similar to that of ME/CFS.

## Figures and Tables

**Figure 1 healthcare-10-02058-f001:**
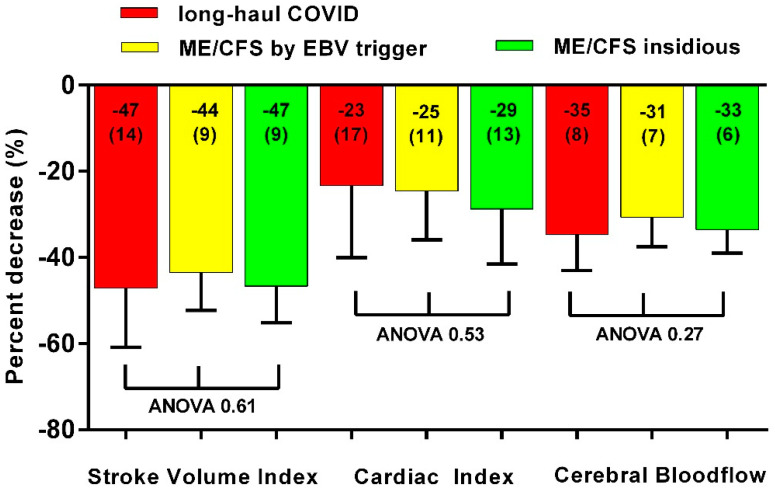
Decrease in stroke volume index, cardiac index and cerebral blood flow in long-haul COVID cases (red bars), in ME/CFS controls triggered by EBV (yellow bars) and in ME/CFS controls with an insidious onset (green bars). ME/CFS: myalgic encephalomyelitis/chronic fatigue syndrome; EBV: Ebstein Barr Virus.

**Figure 2 healthcare-10-02058-f002:**
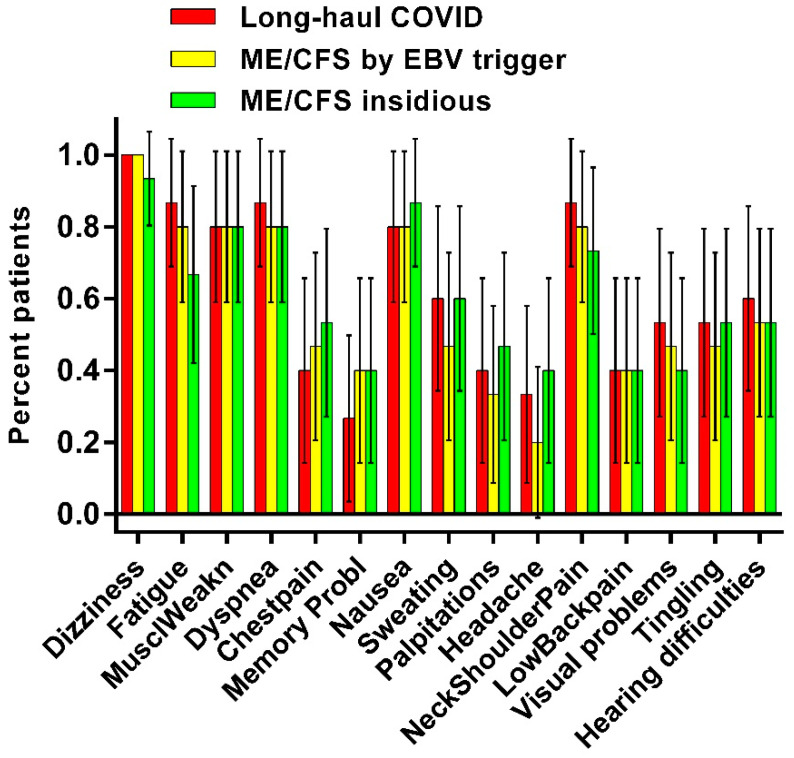
Frequencies of orthostatic intolerance complaints after tilting, as assessed by a questionnaire, in long-haul COVID cases (red), in ME/CFS controls triggered by EBV (yellow) and in ME/CFS controls with an insidious onset (green).

**Figure 3 healthcare-10-02058-f003:**
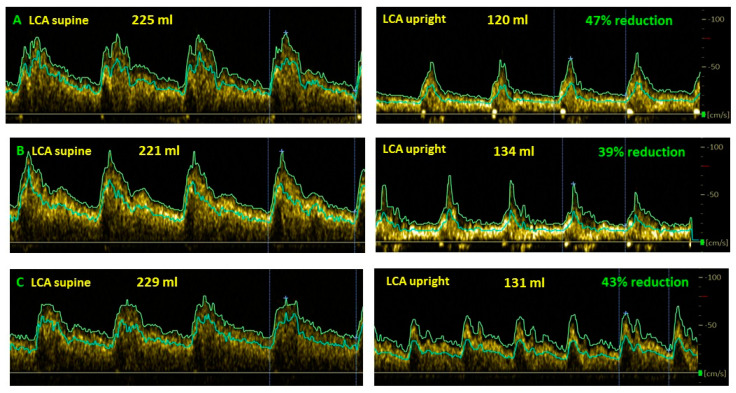
Examples of cerebral blood flow decrease while upright compared to supine in long-haul COVID patients. (**A**): female, 37 years old; (**B**): female, 35 years; (**C**): female, 24 years, LCA: left carotid artery.

**Figure 4 healthcare-10-02058-f004:**
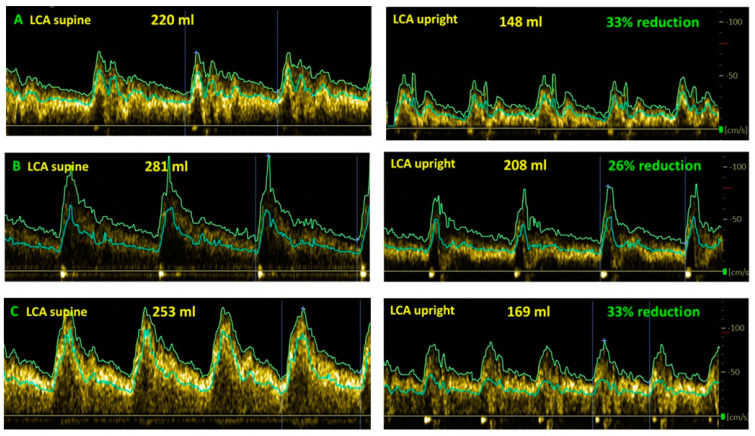
Examples of cerebral blood flow decrease while upright compared to supine in ME/CFS patients with a trigger of EBV. (**A**): female, 38 years; (**B**): female, 21 years old; (**C**): female, 23 years LCA: left carotid artery.

**Figure 5 healthcare-10-02058-f005:**
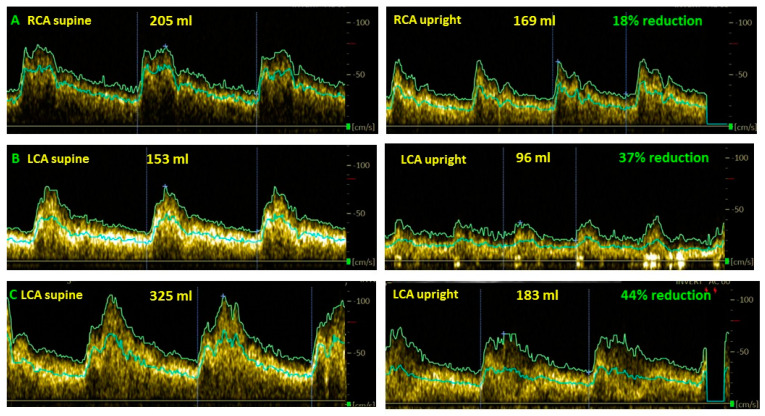
Examples of cerebral blood flow decrease while upright compared to supine in ME/CFS patients with an insidious onset. (**A**): female, 26 years; (**B**): female, 53 years old; (**C**): female, 32 years LCA: left carotid artery; RCA: right carotid artery.

**Table 1 healthcare-10-02058-t001:** Clinical characteristics of long-haul COVID patients (group 1), of ME/CFS patients with Ebstein Barr virus as a trigger (group 2) and of ME/CFS patients with an insidious onset of the disease (group 3).

	Group 1 (*n* = 14)	Group 2 (*n* = 14)	Group 3 (*n* = 14)	*p*-Value
Male/female *	2/12 (14/86%)	2/12 (14/86%)	2/12 (14/86%)	1.0
Age (years)	34 (10)	34 (10)	34 (10)	F (2, 42) = 0.0006; *p* = 0.99
Height (cm)	175 (10)	174 (10)	175 (8)	F (2, 42) = 0.11; *p* = 0.89
Weight (kg)	72 (15)	68 (17)	76 (23)	F (2, 42) = 0.70; *p* = 0.50
BMI (kg/m^2^)	23.5 (4.8)	22.3 (4.5)	24.7 (6.4)	F (2, 42) = 0.76; *p* = 0.47
BSA (m^2^)	1.87 (0.20)	1.81 (0.24)	1.90 (0.28)	F (2, 42) = 0.52; *p* = 0.60
Disease duration (years) #	1 (1–2)	16 (9–23)	11 (4–16)	X^2^(3) = 29.25; *p* < 0.0001. Post-hoc tests: 1 vs. 2 *p* < 0.0001; 1 vs. 3 *p* = 0.0001
Disease severity * ^&^ (mild/moderate/severe)	0/11/3 (0/85/15%)	2/10/2 (14/82/14%)	2/7/5 (14/50/36%)	0.36
OI in daily life yes/no *	14/0 (0/100%)	14/0 (0/100%)	14/0 (0/100%)	1.0

*p*-values: Chi-square analysis (3 × 2 or 3 × 3 table) (*), ordinary one-way ANOVA or Kruskal–Wallis test (#) where appropriate. BMI: body mass index; BSA: body surface area (formula duBois); POTS: postural orthostatic tachycardia syndrome; normHRBP: normal heart rate and blood pressure response; OI: orthostatic intolerance; # Median (IQR); ^&^: severity grading according to the ME criteria [18].

**Table 2 healthcare-10-02058-t002:** Hemodynamic results of the tilt test of long-haul COVID patients (group 1), of ME/CFS patients with Ebstein Barr virus as trigger (group 2) and of ME/CFS patients with an insidious onset of the disease (group 3).

	Group 1 (*n* = 14)	Group 2 (*n* = 14)	Group 3 (*n* = 14)	*p*-Value
Hemodynamics: (normHRBP/POTS)	0/14 (0/100%)	8/6 (57/43%)	7/7 (50/50%)	0.002
HR supine (bpm)	74 (14)	69 (11)	73 (8)	F (2, 42) = 0.93; *p* = 0.40
HR end-tilt (bpm)	108 (13)	95 (20)	103 (20)	F (2, 42) = 1.96; *p* = 0.15
SBP supine (mmHg)	131 (17)	133 (15)	133 (16)	F (2, 42) = 0.059; *p* = 0.94
SBP end-tilt (mmHg)	127 (21)	126 (15)	127 (19)	F (2, 42) = 0.04; *p* = 0.96
DBP supine (mmHg)	80 (13)	77 (9)	82 (18)	F (2, 42) = 0.59; *p* = 0.56
DBP end-tilt (mmHg)	88 (17)	82 (10)	90 (21)	F (2, 42) = 0.85; *p* = 0.43
SVI supine (ml/m^2^)	40 (6)	42 (8)	40 (7)	F (2, 42) = 0.54; *p* = 0.58
SVI end-tilt (ml/m^2^)	21 (5)	23 (5)	21 (7)	F (2, 42) = 0.96; *p* = 0.39
CI supine (L/min/m^2^)	2.91 (0.51)	2.84 (0.52)	2.74 (0.44)	F (2, 42) = 0.44; *p* = 0.65
CI end-tilt (L/min/m^2^)	2.27 (0.75)	2.13 (0.42)	1.94 (0.36)	F (2, 42) = 1.43; *p* = 0.25
P_ET_CO_2_ supine (mmHg)	39 (3)	38 (3)	37 (2)	F (2, 42) = 3.08; *p* = 0.06
P_ET_CO_2_ end-tilt (mmHg)	28 (4)	29 (5)	27 (4)	F (2, 42) = 1.05; *p* = 0.36
CBF supine (ml/min)	620 (68)	618 (75)	631 (87)	F (2, 42) = 0.12; *p* = 0.89
CBF end-tilt (ml/min)	405 (70)	428 (66)	419 (67)	F (2, 42) = 0.47; *p* = 0.63

*p*-values: Chi-square analysis (3 × 2 table) and ordinary one-way ANOVA. HR: heart rate; bpm: beats per minute; SBP: systolic blood pressure; DBD: diastolic blood pressure; SVI: stroke volume index; CI: cardiac index; CB: cerebral blood flow; normHR/BP: patients with a normal heart rate and blood pressure response during the tilt test; POTS; postural orthostatic intolerance syndrome. Upright measurements are the measurements being made just before tilting back. A *p*-value of <0.01 is considered significant.

## Data Availability

Not applicable.

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
