# Peer review of "Orthostatic Intolerance in Long-Haul COVID after SARS-CoV-2: A Case-Control Comparison with Post-EBV and Insidious-Onset Myalgic Encephalomyelitis/Chronic Fatigue Syndrome Patients"

_healthcare, 2022, doi:10.3390/healthcare10102058_

Round 1

Reviewer 1 Report

1. Please describe the patient background in detail. -Comorbidities. -Severity of acute COVID-19.
-Vaccination history
-Treatment medications.
Group 2 and 3, in particular, have been affected for a longer period of time and I am concerned about the type of patient background.   2. Please explain the phenomenon of "fade out" in detail in your discussion section with appropriate references.     You mentioned, "Table 2 shows the hemodynamic results of the tilt table test. POTS was observed in all 14 long-haul COVID patients, 6 patients had POTS and 8 patients showed a normal heart rate and blood pressure pattern in the ME/CFS patients with the EBV trigger, and in the ME/CFS group with an insidious onset 7 patients had POTS and 7 patients showed a normal heart rate and blood pressure pattern (p=0.002). "   I considered that all long-haul COVID patients showed POTS because the disease duration was shorter. Then, you stated in the discussion below. ”In the current study, there was a significant difference in disease duration: median 1 year in the long-haul COVID versus median 11-16 years in the ME/CFS patient groups. One possible explanation for the different prevalence of POTS in long-haul COVID and ME/CFS patients is that with increasing disease duration, the POTS response can "fade out" and give way to a normal heart rate and blood pressure course. ”

Author Response

see the attached document

Reviewer 2 Report

I read this paper with great interest. I thought that ME/CFS and Long COVID are similar pathologies, so this is a convincing conclusion. Hopefully, it would have been even better if it showed how ME/CFS patients with EBV and group 3 have been treated. For many years they have been plagued with illness.

It may be difficult, but I would like to wait for the author's comment that long COVOD may (or may not) be a long-term treatment as well.

Best regards

Author Response

see attached document

Reviewer 3 Report

Dear authors,
I commend you on an excellent paper. I was pleased at the end that you noted that the orthostatic changes in pulse and blood pressure start to decline after five years of illness, because that was one of the only two concerns I had going through the study. And you addressed that well.

My other concern is with how you defined the Long Haul Covid group. A severe problem is that all too many researchers are defining long Covid as ANY persistent symptoms after three months. Whether the symptoms are loss of smell, a severe CVA, or massive MI. Or long Covid. This lumping of all the groups together is problematic. 

So it would be helpful if you can define how you determined that the long-haul Covid group is truly that subset that reflects the ME/CFS/hypothalamic dysfunction group. You did so adequately by noting that no other causes of the illness was found. But if not too difficult, it might be worth noting more specifically that "those with clear organ damage from the Covid such as ARDS, MI, or CVA were excluded." But because figure 2 notes what percent of the long Covid group had fatigue, cognitive dysfunction, widespread pain, disordered sleep, or other symptoms which adequately distinguishes this group from other persistent Covid symptoms, I leave this thought as a consideration rather than a requirement.

But I feel like I am nitpicking here. Bottom line? An excellent job!

Author Response

see attached document
